# *Just Ask for Calibration*: Strategies for Eliciting Calibrated Confidence Scores from Language Models Fine-Tuned with Human Feedback

**Katherine Tian,**[*][†] **Eric Mitchell,**[*][‡] **Allan Zhou,**[‡] **Archit Sharma,**[‡] **Rafael Rafailov**[‡]
**Huaxiu Yao,**[‡] **Chelsea Finn,**[‡] **Christopher D. Manning**[‡]

[†]Harvard University   [‡]Stanford University
ktian@college.harvard.edu
eric.mitchell@cs.stanford.edu

## Abstract

A trustworthy real-world prediction system should produce *well-calibrated* confidence scores; that is, its confidence in an answer should be indicative of the likelihood that the answer is correct, enabling deferral to an expert in cases of low-confidence predictions. Recent studies have shown that unsupervised pre-training produces large language models (LMs) whose conditional probabilities are remarkably well-calibrated. However, the most widely-used LMs are fine-tuned with reinforcement learning from human feedback (RLHF-LMs), and some studies have suggested that RLHF-LMs produce conditional probabilities that are very poorly calibrated. In light of this perceived weakness, we conduct a broad evaluation of methods for extracting confidence scores from RLHF-LMs. For RLHF-LMs such as ChatGPT, GPT-4, and Claude, we find that *verbalized* confidences emitted as output tokens are typically better-calibrated than the model's conditional probabilities on the TriviaQA, SciQ, and TruthfulQA benchmarks, often reducing the expected calibration error by a relative 50%.

## 1 Introduction

Real-world prediction systems invariably make errors. However, some mitigation of these errors is possible if the system produces *well-calibrated*[1] confidence estimates. In this case, the system's least confident predictions correspond to those that are most likely to be incorrect, potentially allowing these predictions to be skipped or overridden by a human. In the context of language models, one consequence of poor calibration may be *hallucination*, where a language model confidently asserts incorrect facts or reasoning. While the ability of very large LMs to absorb and synthesize knowledge about the outside world has gained significant

---
[*]Equal contribution.

[1]i.e., the confidence in a prediction accurately reflects the probability that the prediction is correct (Guo et al., 2017).

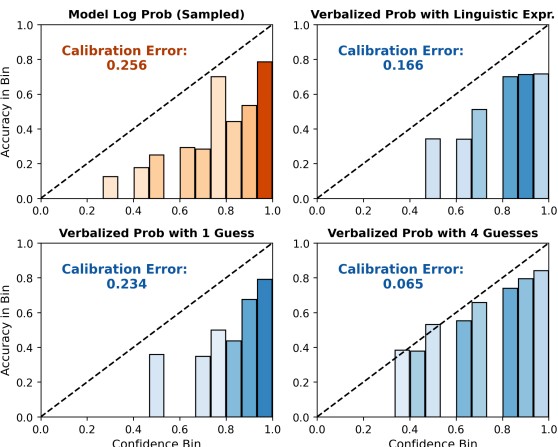

Figure 1: **Verbalized confidence scores (blue) are better-calibrated than log probabilities (orange) for** `gpt-3.5-turbo`. Raw model probabilities (**top-left**) are consistently over-confident. Verbalized numerical probabilities (**bottom**) are better-calibrated. Considering more answer choices (**bottom-right**) further improves verbalized calibration (as in 'Considering the Opposite' in psychology; Lord et al. (1985)). Verbalized *expressions of likelihood* (**top-right**) also provide improved calibration. Bar height is average accuracy of predictions in bin. Darker bars mean more predictions fall in that confidence range. Results computed on SciQ.

attention (Brown et al., 2020; Roberts et al., 2020; Bubeck et al., 2023), relatively little attention has been given to their well-calibratedness (Kadavath et al., 2022). Further, most existing analyses of the calibratedness of LLMs focus on models trained with maximum likelihood, while in practice, the most widely-used LLMs (such as ChatGPT) are fine-tuned using methods such as reinforcement learning from human feedback (Christiano et al., 2017). Some findings suggest that RLHF-LMs may sacrifice well-calibrated predictions for the sake of closer adherence to user instructions in dialogue (Kadavath et al., 2022; OpenAI, 2023), as the reinforcement learning objective encourages the model to allocate probability mass to the most preferred answer(s), rather than matching the relative frequency of possible answers.

This paper evaluates several methods for extracting confidences about model predictions from

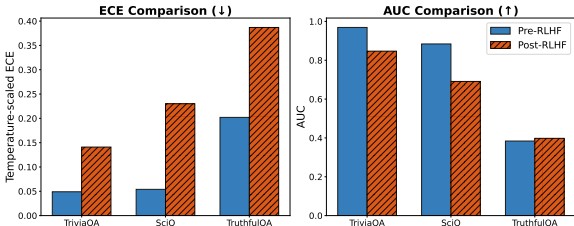

Figure 2: **RLHF generally worsens the calibration of Llama-70B's log probabilities**, as measured by ECE (lower is better) or AUC (higher is better). However, this paper (Tables 1-5) will show that for several strong RLHF-LMs, the model's *verbalized* confidence is often better-calibrated than its log probabilities, reversing some of this degradation. This reversal is strongest for TruthfulQA, an adversarial dataset testing common misconceptions and other difficult queries.

**RLHF-LMs.** Due to concerns that RLHF may cause systematic overconfidence in the model's probabilities (Figure 2), as well as the general unavailability of per-token log-probabilities in widely used RLHF-LMs, we pay particular attention to prompts that elicit *verbalized* probabilities, i.e., the model expresses its confidence in token-space, as either numerical probabilities or another linguistic expression of uncertainty. We find that, surprisingly, popular RLHF-LMs are able to directly verbalize confidence scores that are better-calibrated than the model's conditional probabilities (estimated via sampling), without *any* fine-tuning to learn verbalization. To further improve calibration, we take inspiration from research in human psychology showing that overconfidence can be mitigated by considering alternative answers before responding (Lord et al., 1985; Mussweiler et al., 2000). We show that prompting a model to produce several answer choices before giving its confidence scores significantly improves calibration of verbalized probabilities. Combined with temperature scaling (Guo et al., 2017), this approach generally provides better calibration than model probabilities for ChatGPT[2], GPT-4[3], and Claude 2[4] across three datasets, often reducing expected calibration error (ECE) by over 50%.

**Related Work.** Several studies have examined the calibration of large LMs (Lin et al., 2022a; Park and Caragea, 2022; Kadavath et al., 2022; Xiao et al., 2022; Kuhn et al., 2023), finding that combining large pre-trained LMs with temperature scaling (Guo et al., 2017) produces very well-

calibrated predictions (Kadavath et al., 2022; Xiao et al., 2022; Kuhn et al., 2023). Other work focuses on the tendency of language and dialogue models to use linguistic expressions of uncertainty in a well-calibrated manner (Zhou et al., 2023; Mielke et al., 2022). However, existing studies focus on LMs trained purely with unsupervised learning (although Kadavath et al. (2022) briefly examine RLHF-LMs), while widely used models in practice are fine-tuned with instruction-tuning or RLHF (Christiano et al., 2017). RLHF has been shown to effectively leverage annotations of human preferences to control sentiment (Ziegler et al., 2020), improve summarization or instruction-following quality (Stiennon et al., 2022; Ouyang et al., 2022), and inject behavioral priors of harmlessness (Bai et al., 2022b,a). However, recent work has raised the question of whether or not RLHF harms calibration (OpenAI, 2023). Our work is the first to show that verbalized probabilities are often better-calibrated than the model's conditional probabilities for RLHF-LMs such as ChatGPT, GPT-4, and Claude, and Llama-2-70B-Chat.

## 2 Evaluating Calibration in RLHF-LMs

To study the calibration of RLHF-LMs, we conduct experiments with `gpt-3.5-turbo` (ChatGPT), `gpt-4` (GPT-4), `claude-1` (Claude 1), `claude-2` (Claude 2), and `Llama-2-70b-chat` (Llama-2-70B-Chat).

**Metrics.** We measure calibration with multiple metrics. To measure **ECE** (expected calibration error; Guo et al. (2017)), we bin model predictions by their confidence and measure the average accuracy of predictions in each confidence bin. The ECE is defined as the average (squared) error between the average accuracy and confidence within each bin, where each error is weighted by the fraction of samples falling within the bin. We report raw ECE as well as ECE with temperature scaling (**ECE-t**). Temperature scaling fits a single temperature value $\beta$ to the model's confidences to minimize negative log likelihood (NLL) on the data, giving scaled probability $\tilde{p}_i$ of class $i$ as $\tilde{p}_i \propto p_i^{\beta}$. See Figure 1 for a depiction of ECE binning. Although ECE is a standard and interpretable measure of calibration error, it completely fails to capture the confidences' discriminative power.[5] We therefore also report

---

[2]`gpt-3.5-turbo`, accessed in June 2023.

[3]https://cdn.openai.com/papers/gpt-4-system-card.pdf

[4]https://www.files.anthropic.com/production/images/Model-Card-Claude-2.pdf

[5]For binary classification, a system that guesses randomly and outputs 50% confidence each time has perfect ECE.

| Method | TriviaQA | | | | SciQ | | | | TruthfulQA | | | |
|---|---|---|---|---|---|---|---|---|---|---|---|---|
| | ECE ↓ | ECE-t ↓ | BS-t ↓ | AUC ↑ | ECE ↓ | ECE-t ↓ | BS-t ↓ | AUC ↑ | ECE ↓ | ECE-t ↓ | BS-t ↓ | AUC ↑ |
| Label prob. | 0.140 | 0.097 | 0.142 | 0.869 | 0.256 | 0.180 | 0.223 | 0.752 | 0.451 | 0.317 | 0.345 | 0.418 |
| 'Is True' prob. | 0.164 | 0.159 | 0.165 | 0.826 | 0.312 | 0.309 | 0.309 | 0.677 | 0.470 | 0.471 | 0.476 | 0.384 |
| Entropy | — | — | — | 0.547 | — | — | — | 0.483 | — | — | — | 0.236 |
| Verb. 1S top-1 | 0.068 | 0.076 | 0.138 | 0.879 | 0.234 | 0.084 | 0.214 | 0.744 | 0.389 | 0.256 | 0.322 | 0.545 |
| Verb. 1S top-2 | 0.050 | 0.053 | 0.139 | 0.894 | 0.132 | 0.050 | 0.201 | 0.766 | 0.361 | 0.115 | 0.252 | 0.485 |
| Verb. 1S top-4 | 0.054 | 0.057 | 0.144 | 0.896 | 0.065 | 0.051 | 0.209 | 0.763 | 0.203 | 0.189 | 0.284 | 0.455 |
| Verb. 2S CoT | 0.110 | 0.123 | 0.168 | 0.830 | 0.323 | 0.246 | 0.296 | 0.683 | 0.419 | 0.259 | 0.292 | 0.551 |
| Verb. 2S top-1 | 0.131 | 0.099 | 0.148 | 0.855 | 0.340 | 0.203 | 0.268 | 0.677 | 0.431 | 0.245 | 0.282 | 0.483 |
| Verb. 2S top-2 | 0.047 | 0.045 | 0.147 | 0.887 | 0.169 | 0.040 | 0.201 | 0.768 | 0.395 | 0.101 | 0.224 | 0.517 |
| Verb. 2S top-4 | 0.050 | 0.051 | 0.156 | 0.861 | 0.130 | 0.046 | 0.211 | 0.729 | 0.270 | 0.156 | 0.246 | 0.463 |
| Ling. 1S human | 0.062 | 0.069 | 0.137 | 0.884 | 0.166 | 0.087 | 0.223 | 0.703 | 0.306 | 0.296 | 0.333 | 0.503 |
| Ling. 1S-opt. | 0.058 | 0.066 | 0.135 | 0.878 | 0.064 | 0.068 | 0.220 | 0.674 | 0.125 | 0.165 | 0.270 | 0.492 |

Table 1: Measuring calibration of various methods for extracting confidences from `gpt-3.5-turbo` (ChatGPT). The model's conditional probabilities are relatively poorly calibrated, whether using the model's conditional probability of the label given the query (**Label prob.**) or the probability assigned to 'True' given the query, proposed answer, and a prompt asking if the answer is correct (**'Is True' prob.**). Surprisingly, directly verbalizing a probability (**Verb. 1S** and **Verb. 2S**) or an expression of confidence such as 'highly likely' (**Ling. 1S**) yields *significantly* better-calibrated confidence estimates. 1S refers to one-stage prediction, where the model provides an answer and confidence probability/expression together. 2S refers to two-stage prediction, where the model first gives only an answer, and then in a second stage a confidence. To color the table cells, for each column, we demean and scale by a constant to obtain a shade in [-1,1], where cyan indicates better and orange worse performance.

| Method | TriviaQA | | | | SciQ | | | | TruthfulQA | | | |
|---|---|---|---|---|---|---|---|---|---|---|---|---|
| | ECE ↓ | ECE-t ↓ | BS-t ↓ | AUC ↑ | ECE ↓ | ECE-t ↓ | BS-t ↓ | AUC ↑ | ECE ↓ | ECE-t ↓ | BS-t ↓ | AUC ↑ |
| Label prob. | 0.078 | 0.067 | 0.077 | 0.950 | 0.219 | 0.165 | 0.186 | 0.820 | 0.445 | 0.334 | 0.362 | 0.462 |
| Verb. 1S top-1 | 0.024 | 0.038 | 0.084 | 0.937 | 0.201 | 0.084 | 0.165 | 0.843 | 0.350 | 0.156 | 0.227 | 0.622 |
| Verb. 1S top-2 | 0.025 | 0.034 | 0.084 | 0.949 | 0.140 | 0.048 | 0.185 | 0.813 | 0.315 | 0.112 | 0.228 | 0.623 |
| Verb. 1S top-4 | 0.041 | 0.039 | 0.081 | 0.959 | 0.056 | 0.059 | 0.185 | 0.815 | 0.198 | 0.144 | 0.245 | 0.619 |
| Ling. 1S-human | 0.051 | 0.041 | 0.086 | 0.931 | 0.148 | 0.024 | 0.170 | 0.835 | 0.241 | 0.151 | 0.228 | 0.651 |
| Ling. 1S-opt. | 0.056 | 0.051 | 0.088 | 0.927 | 0.028 | 0.052 | 0.172 | 0.828 | 0.082 | 0.105 | 0.212 | 0.632 |

Table 2: `gpt-4`'s verbalized probabilities are substantially better-calibrated than the model probabilities themselves, even after temperature scaling, similarly to `gpt-3.5-turbo` in Table 1.

Brier Score (BS; Brier (1950)) on temperature-scaled confidences (**BS-t**), a proper scoring rule (Ovadia et al., 2019) that is the mean squared error between the confidences and the correctness labels. Finally, we assess calibration using a metric from the selective classification literature (Geifman and El-Yaniv, 2017), specifically, the area under the curve of selective accuracy and coverage (**AUC**).

**Datasets.** Our experiments use three question-answering datasets assessing factual knowledge. TriviaQA (Joshi et al., 2017) contains 650k question-answer pairs gathered by trivia enthusiasts; SciQ (Welbl et al., 2017) contains approximately 14k crowdsourced science exam question-answer pairs; TruthfulQA (Lin et al., 2022b) contains 817 questions designed to test language models' tendency to mimic human falsehoods. We sample 1000 questions from the validation split of TriviaQA (`rc.web.nocontext`) and SciQ and all 817 questions from the validation split of TruthfulQA (`generation`) for our experiments.

**Evaluation protocol.** For each dataset, we generate a response and corresponding confidence from each method on each of the evaluation questions. Because calibration essentially quantifies the relationship between model confidence and correctness, computing correctness is crucial to accurate measurements of calibration. However, we find doing so to be a challenge, especially in datasets where only a single ground-truth answer (but not aliases or semantically equivalent rephrases) is provided. To avoid excessive false negatives in our correctness computation as a result of exact-match evaluation, we use either GPT-4 or GPT-3.5 to evaluate whether a response is essentially equivalent to the ground truth answer; see Appendix C for the complete equivalence-checking procedure.

**Methods.** We compare a wide variety of methods for extracting confidence estimates from LLMs. For a comprehensive list of the prompts used for each method, see Appendix Table 6.

First, we consider two methods that leverage the true conditional distribution of the model to gener-

| | **TriviaQA** | | | | **SciQ** | | | | **TruthfulQA** | | | |
|---|---|---|---|---|---|---|---|---|---|---|---|---|
| **Method** | ECE ↓ | ECE-t ↓ | BS-t ↓ | AUC ↑ | ECE ↓ | ECE-t ↓ | BS-t ↓ | AUC ↑ | ECE ↓ | ECE-t ↓ | BS-t ↓ | AUC ↑ |
| Label prob. | 0.074 | 0.079 | 0.117 | 0.915 | 0.216 | 0.149 | 0.195 | 0.786 | 0.432 | 0.304 | 0.335 | 0.418 |
| Verb. 1S top-1 | 0.049 | 0.059 | 0.160 | 0.839 | 0.265 | 0.103 | 0.247 | 0.663 | 0.440 | 0.134 | 0.204 | 0.411 |
| Verb. 1S top-2 | 0.046 | 0.047 | 0.158 | 0.875 | 0.207 | 0.040 | 0.225 | 0.693 | 0.450 | 0.085 | 0.197 | 0.409 |
| Verb. 1S top-4 | 0.075 | 0.079 | 0.176 | 0.814 | 0.151 | 0.057 | 0.226 | 0.667 | 0.372 | 0.105 | 0.183 | 0.377 |
| Ling. 1S human | 0.053 | 0.050 | 0.151 | 0.867 | 0.253 | 0.118 | 0.245 | 0.664 | 0.443 | 0.358 | 0.340 | 0.384 |
| Ling. 1S-opt. | 0.074 | 0.060 | 0.149 | 0.863 | 0.089 | 0.082 | 0.238 | 0.623 | 0.139 | 0.148 | 0.228 | 0.350 |

Table 3: Claude-1 produces similar- or better-calibrated log probabilities to `gpt-3.5-turbo`, but is less able to verbalize well-calibrated confidences, compared to models in the GPT family of RLHF-LMs. Claude-1 has since been deprecated.

| | **TriviaQA** | | | | **SciQ** | | | | **TruthfulQA** | | | |
|---|---|---|---|---|---|---|---|---|---|---|---|---|
| **Method** | ECE ↓ | ECE-t ↓ | BS-t ↓ | AUC ↑ | ECE ↓ | ECE-t ↓ | BS-t ↓ | AUC ↑ | ECE ↓ | ECE-t ↓ | BS-t ↓ | AUC ↑ |
| Label prob. | 0.089 | 0.089 | 0.137 | 0.882 | 0.181 | 0.176 | 0.237 | 0.762 | 0.409 | 0.368 | 0.405 | 0.319 |
| Verb. 1S top-1 | 0.072 | 0.071 | 0.141 | 0.903 | 0.204 | 0.054 | 0.201 | 0.776 | 0.345 | 0.115 | 0.215 | 0.573 |
| Verb. 1S top-2 | 0.049 | 0.054 | 0.133 | 0.918 | 0.134 | 0.041 | 0.211 | 0.754 | 0.359 | 0.085 | 0.223 | 0.491 |
| Verb. 1S top-4 | 0.072 | 0.063 | 0.158 | 0.890 | 0.048 | 0.052 | 0.216 | 0.711 | 0.274 | 0.075 | 0.208 | 0.473 |
| Ling. 1S human | 0.085 | 0.061 | 0.151 | 0.878 | 0.238 | 0.026 | 0.209 | 0.756 | 0.381 | 0.242 | 0.305 | 0.530 |
| Ling. 1S-opt. | 0.060 | 0.070 | 0.151 | 0.874 | 0.049 | 0.056 | 0.214 | 0.738 | 0.099 | 0.130 | 0.266 | 0.446 |

Table 4: Claude-2 has weaker conditional probabilities than Claude-1 and GPT-*, but its verbalized calibration provides consistent improvement over conditional probabilities at a level comparable to GPT-3.5 and surpassing GPT-* on TruthfulQA.

ate confidence scores. The simplest is **Label prob.**, which uses the conditional probability distribution $p(y|x)$ of the model given a question $x$, which we estimate using $n = 10$ samples, since many RLHF-LMs are closed-source and do not offer per-token probabilities.[6][7] We return the most common answer, using the LLM-based equivalence function to determine when two lexically different answers are semantically equivalent. In a variation of the method described by Kadavath et al. (2022) (again, we use samples since model probabilities are not available), **'Is True' prob.** samples a single answer $\hat{y}$ from the model given a question $x$, and the probability it is true is estimated by the probability the model assigns to 'True' when asked if the given answer is true (where once again the probabilities are estimated via samples), i.e., $p(\text{True}|x, \hat{y})$.

Next, we consider methods that extract confidence scores through *verbalization* (Lin et al., 2022a), i.e., where the model expresses its confidence in token space, either with numerical probabilities or linguistic expressions of likelihood.[8] First, **Verb. 1S top-$k$** prompts the model to produce $k$ guesses and a probability that each is correct all in a single response (i.e., '1 stage'). We take the highest-probability prediction and its as-

sociated probability as the model's output and confidence. **Verb. 2S top-$k$** similarly uses numerical probabilities, except the model is first asked to provide only its answers, and afterwards, in a second round of dialogue, asked to assign probabilities of correctness to each answer (i.e., '2 stages'). **Verb. 2S CoT** uses a chain-of-thought prompt before giving a single answer, and in a second round of dialogue, the model is prompted to assign a probability to that answer (with the chain of thought present in the model's context). **Ling. 1S-human** uses *linguistic* likelihood expressions, rather than numerical probabilities, to express uncertainty. The model is prompted to assign confidences to its guesses by choosing from a set of linguistic expressions of uncertainty: {Almost certain, Likely, ..., Almost no chance}. Each linguistic likelihood expression is mapped to a probability using responses from a human survey on social media with 123 respondents (Fagen-Ulmschneider, 2023). **Ling. 1S-opt.** uses a held out set of calibration questions and answers to compute the average accuracy for each likelihood expression, using these 'optimized' values instead. Expressions that are not used for at least $\frac{1}{N}$ of questions, where $N$ is the number of calibration questions, simply use the human probability.

## 3   Results

Tables 1–5 show the results of evaluating various methods for extracting confidence from RLHF-LMs on `gpt-3.5-turbo`, `gpt-4`, `claude-1`,

---

[6]We evaluated `gpt-3.5-turbo` on all three datasets using $n = 20$ samples, but the calibration did not meaningfully improve, so we always use $n = 10$ to reduce API costs.

[7]For each closed LM, we use its default sampling parameters (top-p 1.0 for GPT-* and top-p 0.7 for Claude). For Llama-2, we use temperature 1.0 and top-p 1.0.

[8]However, note that *none* of the methods described fine-tune the model to perform better on verbalization.

| Method | TriviaQA | | | | SciQ | | | | TruthfulQA | | | |
|---|---|---|---|---|---|---|---|---|---|---|---|---|
| | ECE ↓ | ECE-t ↓ | BS-t ↓ | AUC ↑ | ECE ↓ | ECE-t ↓ | BS-t ↓ | AUC ↑ | ECE ↓ | ECE-t ↓ | BS-t ↓ | AUC ↑ |
| Label prob. | 0.151 | 0.124 | 0.156 | 0.865 | 0.266 | 0.189 | 0.243 | 0.707 | 0.405 | 0.361 | 0.396 | 0.407 |
| Verb. 1S top-1 | 0.071 | 0.067 | 0.186 | 0.793 | 0.196 | 0.053 | 0.239 | 0.648 | 0.386 | 0.172 | 0.266 | 0.502 |
| Verb. 1S top-2 | 0.060 | 0.073 | 0.194 | 0.815 | 0.153 | 0.032 | 0.230 | 0.667 | 0.340 | 0.037 | 0.227 | 0.440 |
| Verb. 1S top-4 | 0.069 | 0.079 | 0.182 | 0.816 | 0.105 | 0.043 | 0.229 | 0.648 | 0.231 | 0.102 | 0.237 | 0.465 |
| Ling. 1S human | 0.179 | 0.115 | 0.195 | 0.749 | 0.071 | 0.101 | 0.252 | 0.603 | 0.376 | 0.366 | 0.383 | 0.407 |
| Ling. 1S-opt. | 0.077 | 0.068 | 0.186 | 0.779 | 0.019 | 0.042 | 0.236 | 0.590 | 0.047 | 0.051 | 0.239 | 0.435 |

Table 5: With Llama2-70B-Chat, verbalized calibration provides improvement over conditional probabilities across some metrics, but the improvement is much less consistent compared to GPT-* and Claude-*.

`claude-2`, and `Llama-2-70b-chat`, respectively. We distill several key conclusions from these experiments. **1. Large RLHF-LMs can often directly verbalize better-calibrated confidences (either a numerical confidence probability or an expression such as 'highly likely') than the models' conditional probabilities. 2.** Among the methods for verbalizing probabilities directly, we observe that generating and evaluating multiple hypotheses improves calibration (see Figure 1), similarly to humans (Lord et al., 1985), and corroborating a similar finding in LMs (Kadavath et al., 2022). **3.** Language models can express their uncertainty with *numerical probabilities* as well or better than with *words*, which is surprising in light of long-standing difficulties in representing numbers in language models (Thawani et al., 2021). **4.** Chain-of-thought prompting does not improve verbalized calibration (see Appendix Figure 5 for additional CoT results). **5.** The calibration of both Claude models' conditional probabilities roughly falls between `gpt-3.5-turbo` and `gpt-4`; however, while Claude 1 is much weaker at *verbalizing* its confidence, Claude 2 is generally a bit stronger than `gpt-3.5-turbo` at verbalizing. The verbal calibration of the open source model `Llama-2-70b-chat` is generally weaker than that of closed source models but still demonstrates improvement over its conditional probabilities by some metrics, and does so most clearly on TruthfulQA.

## 4 Discussion

In summary, we study the calibration of widely used RLHF-LMs. We first replicate the finding for GPT-4 (OpenAI, 2023) that RLHF can worsen the calibration of a model's conditional probabilities using the open-source Llama-2-70B base and chat models (Figure 2). To mitigate this regression and ease extraction of calibrated confidence scores for models for which log probabilities are not available, we propose and study new methods that can elicit calibrated confidences from RLHF-LMs by prompting the model to *verbalize* its confidence in token space. We find verbalized probabilities are better-calibrated than conditional probabilities across several closed models, with mixed results for Llama-2-70B-Chat.

Our results raise several questions for future work. Most notably, the difference between GPT-*, Claude-*, and Llama-2's ability to verbalize confidence is significant. What factors are important for learning this skill? Additionally, the 1-stage and 2-stage verbalized numerical confidence prompts sometimes differ drastically in the calibration of their confidences. How can we reduce sensitivity of a model's calibration to the prompt? Going beyond question-answering, can we leverage good calibration in short-answer settings to improve the reliability of long-form generations, perhaps by breaking down long-form generation into a sequence of short questions? Finally, to what extent does a language model's calibration depend on the domain; do our conclusions in the context of factual recall hold in the context of reasoning or arithmetic? Answering these questions provides one path toward building more trustworthy and useful language systems.

**Limitations.** While our work demonstrates a promising new approach to generating calibrated confidences through verbalization, there are limitations that could be addressed in future work. First, our experiments are focused on factual recall-oriented problems, and the extent to which our observations would hold for reasoning-heavy settings is an interesting open question. Additionally, the lack of technical details available for many state-of-the-art closed RLHF-LMs may limit our ability to understand what factors enable a model to verbalize well-calibrated confidences and differences in this ability across different models. Finally, our study is limited to short-form question-answering; future work should extend this analysis to longer-form generation settings.

**Acknowledgements.** CF and CDM are CIFAR Fellows. EM gratefully acknowledges funding from a Knight-Hennessy Graduate Fellowship. AZ is supported by the NSF graduate research fellowship program. This research was supported in part by Juniper Networks, Apple, and ONR grant N00014-20-1-2675. The authors thank Yoonho Lee and Noah Goodman for helpful feedback on calibration metrics and experiment design.

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

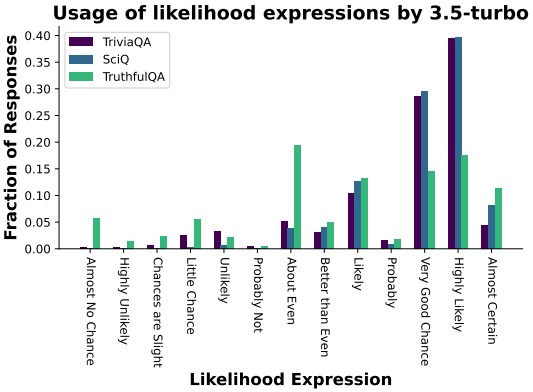

Figure 3: `gpt-3.5-turbo` usage rate of each likelihood expression; the model displays much lower verbalized confidence on TruthfulQA than on standard factual recall problems.

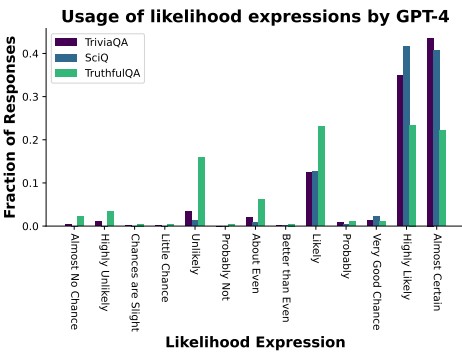

Figure 4: `gpt-4` usage rate of each likelihood expression; the model displays markedly lower verbalized confidence on TruthfulQA than on standard factual recall problems.

## A  Additional Results

Here, we include the likelihood expression usage distribution for `gpt-3.5` and `gpt-4` in Figures 3 and 4, respectively. `gpt-3.5` is systematically less confident for TruthfulQA. The contrast between model confidence for TriviaQA and SciQ compared with TruthfulQA is even more stark for `gpt-4`.

We also provide additional calibration results for chain-of-thought methods. We compare a one-stage verbalized CoT prompt (Verb. 1S CoT), a two-stage verbalized CoT prompt (Verb. 2S CoT), and a two-stage verbalized method that uses CoT just before eliciting the numerical confidence (Verb. 2S Cot Prob) instead of before the guess, as shown for `gpt-3.5` on Trivia QA, SciQ, and Truthful QA in Figure 5. We find that CoT does not noticeably improve calibration across any setting or dataset.

## B  Fitting Procedure for Temperature and Probabilities for Linguistic Expressions

To fit the temperature that is used to compute ECE-t and BS-t we split our total data into 5 folds. For

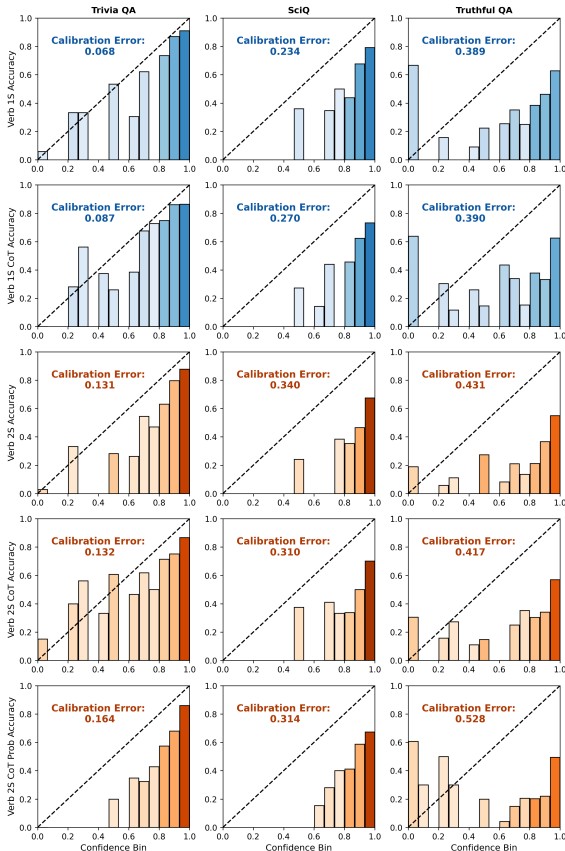

Figure 5: Expected calibration error is not consistently improved for any CoT prompt variant on `gpt-3.5-turbo`.

each fold, we use it once to fit a temperature and evaluate metrics on the remaining folds. We find that fitting the temperature on 20% of the data yields relatively stable temperatures across folds. We report the average temperature-scaled ECE and BS as ECE-t and BS-t.

To compute ECE and AUC for Ling. 1S-opt., we similarly split our total data into 5 folds, using 4 folds to fit the probabilities behind each linguistic expression of confidence, then evaluating on the remaining fold. To compute ECE-t and BS-t for Ling. 1S-opt, we hold out one of the 5 folds to fit temperature. We use 3 folds to fit probabilities for linguistic expressions, compute the temperature based on these probabilities on the temperature set, and evaluate metrics on the last fold. We then average metrics across all 20 rotations of folds.

## C  Prompt Templates

The prompt template for each sampling method is provided in Table 6. The question is substituted for the variable ${THE_QUESTION} in each prompt. To evaluate answer correctness, we use `gpt-3.5-turbo` for SciQ and TruthfulQA and `gpt-4` for TriviaQA due to `gpt-3.5-turbo`'s

| Method | Template |
|--------|----------|
| Label prob. | Provide your best guess for the following question.  Give ONLY the guess, no other words or explanation.\n\nFor example:\n\nGuess: <most likely guess, as short as possible; not a complete sentence, just the guess!>\n\nThe question is:${THE_QUESTION} |
| 'Is True' prob. | Question: ${QUESTION}\nProposed Answer: ${ANSWER}\nIs the proposed answer:\n\t(A) True or\n\t(B) False?\n The proposed answer is: |
| Verb. 1S top-1 | Provide your best guess and the probability that it is correct (0.0 to 1.0) for the following question. Give ONLY the guess and probability, no other words or explanation. For example:\n\nGuess: <most likely guess, as short as possible; not a complete sentence, just the guess!>\n Probability: <the probability between 0.0 and 1.0 that your guess is correct, without any extra commentary whatsoever; just the probability!>\n\nThe question is: ${THE_QUESTION} |
| Verb. 1S top-$k$ | Provide your ${k} best guesses and the probability that each is correct (0.0 to 1.0) for the following question. Give ONLY the guesses and probabilities, no other words or explanation. For example:\n\nG1: <first most likely guess, as short as possible; not a complete sentence, just the guess!>\n\nP1: <the probability between 0.0 and 1.0 that G1 is correct, without any extra commentary whatsoever; just the probability!> ... G${k}: <${k}-th most likely guess, as short as possible; not a complete sentence, just the guess!>\n\nP${k}: <the probability between 0.0 and 1.0 that G${k} is correct, without any extra commentary whatsoever; just the probability!> \n\nThe question is: ${THE_QUESTION} |
| Verb. 2S CoT | Provide your best guess for the following question.  Before giving your answer, provide a step-by-step explanation of your thought process.  Then on a new line give the guess with no other words or explanation.\n\nFor example:\n\nExplanation: <one sentence step-by-step explanation of your thought process>\n\nGuess: <most likely guess, as short as possible; not a complete sentence, just the guess!>\n\nThe question is: ${THE_QUESTION}
Provide the probability that your guess is correct. Give ONLY the probability, no other words or explanation.\n\nFor example:\n\nProbability: <the probability between 0.0 and 1.0 that your guess is correct, without any extra commentary whatsoever; just the probability!>\n |
| Verb. 2S top-1 | Provide your best guess for the following question.  Give ONLY the guess, no other words or explanation.\n\nFor example:\n\nGuess: <most likely guess, as short as possible; not a complete sentence, just the guess!>\n\nThe question is:${THE_QUESTION}
Provide the probability that your guess is correct. Give ONLY the probability, no other words or explanation.\n\nFor example:\n\nProbability: <the probability between 0.0 and 1.0 that your guess is correct, without any extra commentary whatsoever; just the probability!>\n |
| Verb. 2S top-$k$ | Provide your ${k} best guesses for the following question. Give ONLY the guesses, no other words or explanation. For example:\n\nG1: <first most likely guess, as short as possible; not a complete sentence, just the guess!>\n\nP1: <the probability between 0.0 and 1.0 that G1 is correct, without any extra commentary whatsoever; just the probability!> ... G${k}: <${k}-th most likely guess, as short as possible; not a complete sentence, just the guess!>\n\nThe question is:${THE_QUESTION}
Provide the probability that each of your guesses is correct.  Give ONLY the probabilities, no other words or explanation.\n\nFor example:\n\nP1: <the probability between 0.0 and 1.0 that G1 is correct, without any extra commentary whatsoever; just the probability!>\n... P${k}: <the probability between 0.0 and 1.0 that G${k} is correct, without any extra commentary whatsoever; just the probability!> |
| Ling. 1S | Provide your best guess for the following question, and describe how likely it is that your guess is correct as one of the following expressions: ${EXPRESSION_LIST}. Give ONLY the guess and your confidence, no other words or explanation.  For example:\n\nGuess: <most likely guess, as short as possible; not a complete sentence, just the guess!>\nConfidence: <description of confidence, without any extra commentary whatsoever; just a short phrase!>\n\nThe question is: ${THE_QUESTION} |

Table 6: Prompt templates for each method evaluated. Methods above the double line use multiple samples in order to estimate confidence scores; methods below the double line use the verbalized confidences directly, requiring only a single sample.

high disagreement with a human evaluator on TriviaQA. Using the ground truth answer as ${GOLD_ANSWER} and the model-generated answer as ${PRED_ANSWER}, we use the following prompt template:

```
 Are the following two answers to my
question Q semantically equivalent?\n\nQ:
${THE_QUESTION}\nA1:  ${GOLD_ANSWER}\nA2:
${PRED_ANSWER}\n\nPlease  answer  with  a
single word, either "Yes." or "No.", and
explain your reasoning.
```