# OpenReview forum: "Just Ask for Calibration: Strategies for Eliciting Calibrated Confidence Scores from Language Models Fine-Tuned with Human Feedback"
_EMNLP/2023/Conference — EMNLP 2023 Main_

### Official Review · Reviewer_B7ZR · 2023-07-30

**Soundness:** 3

**Excitement:**

3: Ambivalent: It has merits (e.g., it reports state-of-the-art results, the idea is nice), but there are key weaknesses (e.g., it describes incremental work), and it can significantly benefit from another round of revision. However, I won't object to accepting it if my co-reviewers champion it.

**Paper Topic And Main Contributions:**

This paper measures the calibration of various methods for extracting confidences from RLHF-LMs (e.g., ChatGPT, GPT4, and Claude).
They demonstrate that Large RLHF-LMs, particularly GPT-family models, can often directly verbalize better-calibrated confidences.
Besides, generating and evaluating multiple hypotheses improves calibration.


**Questions For The Authors:**

see the reason to reject

**Reasons To Accept:**

Good writing and easy to follow.

An interesting finding of extracting confidence via two-stage prediction can improve the calibration of RLHF-LMs.


**Reasons To Reject:**

The evaluation of three RLHF-LMs appears to lack meaningful conclusions beyond the presentation of two additional tables (i.e., Tables 2 and 3).

I am interested in understanding the reason behind the statement, "The calibration of Claude's conditional probabilities falls between GPT-3.5-turbo and GPT-4, but its ability to verbalize calibrated confidences is weaker than both GPT models." What are the specific differences between the two models, and are there any more generalized and concrete observations?

Are conclusions 2 and 3 of Section 3 applicable to all LMs, or are these phenomena specific only to RLHF-LMs?

**Reproducibility:**

4: Could mostly reproduce the results, but there may be some variation because of sample variance or minor variations in their interpretation of the protocol or method.

**Reviewer Confidence:**

3: Pretty sure, but there's a chance I missed something. Although I have a good feel for this area in general, I did not carefully check the paper's details, e.g., the math, experimental design, or novelty.

---

> ### Author Rebuttal · Authors · 2023-08-29
>
> Thank you for your careful review of our work and belief that it has interesting findings!
>
> We note that in addition to reporting evaluations of multiple widely-used RLHF-LMs, we make the following concrete conclusions of scientific and practical importance:
> * RLHF-LMs can often produce better-calibrated confidences than suggested by widely-cited recent work on RLHF (the "GPT-4 paper") [1] through verbalization
> * Practitioners using these models would benefit from using verbalized probabilities to assess model confidence, which was not previously known. This point is particularly important because using model logits to extract confidence is not an option for the most popular RLHF-LM APIs, so verbalized probabilities are the only choice.
> * The differences in verbalization ability between GPT* models and Claude may be impacted by the decision to perform RLHF (GPT*) or RLAIF (Claude), though it is difficult to verify this hypothesis without closer access to the models
> * To our knowledge, our work is the first to show a phenomenon like "considering the opposite" from human psychology (which is able to reduce human overconfidence) can improve calibration in LLMs (via top-k verbalized methods).
>
> Because most of the training details of ChatGPT, GPT-4, and Claude are unknown, it is difficult to draw additional specific conclusions about the impact of particular training-time design decisions. However, we believe the discovery of the utility of verbalized probabilities without fine-tuning is of significant scientific importance (where does this ability come from?) and practical relevance (practitioners can benefit from using this technique).
>
> Regarding the applicability of points 2 and 3 to non-RLHF-LMs, we note that prior work [1] has shown that non-RLHF-LMs can be fine-tuned to verbalize probabilities. However, no work to our knowledge has shown that LMs can verbalize confidence effectively without any fine-tuning, which we hypothesize is due to the difficulty of constructing a prompt that effectively specifies the intended confidence-verbalization behavior. This situation is complicated by the finding that large pre-trained LMs may almost entirely ignore the labels in their few-shot examples [2], so the numerical confidences in the prompt may be ignored to a non-trivial extent.
>
> Nonetheless, we perform several experiments with the pre-RLHF InstructGPT and pre-RLHF Llama-2 model, and discuss the results below.
>
> * InstructGPT's model probabilities are well-calibrated across TriviaQA, SciQ, and TruthfulQA (ECE-t 0.040, 0.075, 0.216) in comparison to post-RLHF GPT-3.5/ChatGPT (ECE-t 0.097, 0.180, 0.317). However, InstructGPT demonstrates poor verbalized calibration with k=1 (ECE-t 0.221, 0.264, 0.371) in comparison to GPT-3.5 (ECE-t 0.076, 0.084, 0.256) and struggles to express k=4 verbalized calibration at all, while GPT-3.5 benefits from k=4 verbalization (ECE-t 0.057, 0.051, 0.189), outperforming the average ECE of InstructGPT's logprobs.
>
> * To consider the pre-RLHF Llama-2 base model, we use a few shot prompt that provides example answers and their corresponding model conditional probabilities. Similarly to InstructGPT, Llama-2-base has well-calibrated model conditional probabilities across three datasets (ECE-t 0.032, 0.061, 0.228) but has uninformative k=1 verbalized calibration (ECE-t 0.129, 0.245, 0.320).
>
>
> Although future investigation of more complex methods may be able to elicit calibrated verbalized confidences from pre-RLHF models, this ability is not easily found in pre-RLHF models.
>
>
> [1] Teaching models to express their uncertainty in words. Stephanie Lin et al., 2022. TMLR.
>
> [2] Rethinking the Role of Demonstrations: What Makes In-Context Learning Work? Sewon Min et al., 2022. EMNLP.

---

### Official Review · Reviewer_P3om · 2023-08-07

**Soundness:** 4

**Excitement:**

4: Strong: This paper deepens the understanding of some phenomenon or lowers the barriers to an existing research direction.

**Paper Topic And Main Contributions:**

The paper tackles the question of extracting well-calibrated confidence scores from state-of-the-art language models. The main discoveries are

1. LM can directly verbalize confidence scores well
2. Eliciting multiple guesses can further improve the confidence scores


**Reasons To Accept:**

This paper tackles an important and practical problem. Extracting confidence from a black-box LLM is an important direction that deserves more research. The solutions are simple and effective. The ablation experiment is complete.

**Reasons To Reject:**

The trends of ChatGPT and Claude are slightly different. In this case, the paper would be stronger if more language models are analyzed, for example, LLaMA, in which logits can be extracted from the LM.

**Reproducibility:**

4: Could mostly reproduce the results, but there may be some variation because of sample variance or minor variations in their interpretation of the protocol or method.

**Reviewer Confidence:**

4: Quite sure. I tried to check the important points carefully. It's unlikely, though conceivable, that I missed something that should affect my ratings.

---

> ### Author Rebuttal · Authors · 2023-08-29
>
> Thank you for your review - we're glad you appreciated the simplicity of our methods and the completeness of our experiments! We've now performed initial experiments on Llama-2 (which, of course, came out after the submission date), including both the base model and the RLHF-tuned version. The results are below:
>
> First, we verify that RLHF worsens the calibration of model logprobs in LLama-2 (RLHF ECE-t 0.134, 0.207, 0.382 vs. Pretrained ECE-t 0.032, 0.061, 0.228), which follows the same pattern in GPT-4 [1].
>
> Next, comparing Llama-2-RLHF to GPT-3.5 and Claude, we observe that GPT-3.5 has the best k=1 verbalized calibration overall. Llama-2-RLHF has the weakest calibration from model probabilities, and its verbalized calibration is overall comparable to Claude, with worse calibration on TriviaQA and better on TruthfulQA (see table of full results below). We hope that as more large open-source models become available, future work can study the differences between these models and how they lead to the ability to verbalize calibrated confidences.
>
>
> Table. ECE-t, BS-t, AUC on TriviaQA; ECE-t, BS-t, AUC on SciQ; ECE-t, BS-t, AUC on TruthfulQA:
>
> GPT-3.5 verbalized k=1:           0.076, 0.138, 0.879; 0.084, 0.214, 0.744; 0.256, 0.322, 0.545
>
> Claude verbalized k=1:            0.059, 0.160, 0.839; 0.103, 0.247, 0.663; 0.134, 0.204, 0.411
>
> Llama2 verbalized k=1: 0.045, 0.187, 0.779; 0.027, 0.235, 0.663; 0.097, 0.241, 0.475
>
> GPT-3.5 sampling:                   0.097, 0.142, 0.869; 0.180, 0.223, 0.752; 0.317, 0.345, 0.418
>
> Claude sampling:                     0.079, 0.117, 0.915; 0.149, 0.195, 0.786, 0.304, 0.335, 0.418
>
> Llama2 sampling:          0.130, 0.155, 0.847; 0.234, 0.268, 0.690; 0.382, 0.412, 0.398
>
> [1] GPT-4 Technical Report. OpenAI, 2023. arXiv.

---

### Official Review · Reviewer_M6yR · 2023-08-13

**Soundness:** 3

**Excitement:**

3: Ambivalent: It has merits (e.g., it reports state-of-the-art results, the idea is nice), but there are key weaknesses (e.g., it describes incremental work), and it can significantly benefit from another round of revision. However, I won't object to accepting it if my co-reviewers champion it.

**Paper Topic And Main Contributions:**

The paper contributes by addressing the imperative requirement for a trustworthy real-world prediction system with well-calibrated confidence scores. It acknowledges the existing success of unsupervised pre-training in generating highly calibrated conditional probabilities within large language models (LMs). However, recognizing the potential calibration issues within reinforcement learning fine-tuned LMs (RLHF-LMs), the study undertakes an extensive assessment of techniques to derive confidence scores from such models. Notably, for RLHF-LMs like ChatGPT, GPT-4, and Claude, the research demonstrates that the utilization of verbalized confidences as output tokens yields improved calibration compared to the model's native conditional probabilities. These findings are validated across benchmark datasets including TriviaQA, SciQ, and TruthfulQA, resulting in a substantial reduction of the anticipated calibration error by approximately 50%.







**Reasons To Accept:**

The paper introduces novel perspectives on calibration and presents some promising experimental results.

**Reasons To Reject:**

The paper delineates a calibration approach and presents some favorable conclusions; however, the method is relatively simple and in some extent, lacks novelty.

**Reproducibility:**

3: Could reproduce the results with some difficulty. The settings of parameters are underspecified or subjectively determined; the training/evaluation data are not widely available.

**Reviewer Confidence:**

4: Quite sure. I tried to check the important points carefully. It's unlikely, though conceivable, that I missed something that should affect my ratings.

---

> ### Author Rebuttal · Authors · 2023-08-29
>
> Thank you for your feedback - we appreciate that you saw our work as an extensive study providing novel perspectives!
>
> Regarding simplicity of the method:
> * Since fine-tuning widely-used closed models is expensive or impossible and these models are expensive to sample from, simple prompting-based approaches to improving calibration are more likely to see greater adoption and improve real-world model trustworthiness than more complex or expensive algorithms. Additionally, our method does not require having access to model weights and can be used on closed RLHF models.
> * Many users of RLHF-LMs are not AI experts, and so simple methods like our k-shot verbalized probability prompts are more amenable to being used by practitioners.
>
> Regarding novelty:
> * Although our work does not present a complex new methodology, our work is the first to show that LLMs can produce well-calibrated verbalized confidences without any fine-tuning or training specifically for this ability, which we consider to be a non-obvious result that is very relevant to current LLMs. Future research directions could investigate how or why this ability emerges, especially post-RLHF.
> * Beyond showing that top-k verbalized probabilities are quite well-calibrated, a primary scientific contribution of our work is showing that RLHF-LMs can produce better-calibrated confidences than implied by widely-cited recent work on RLHF (the "GPT-4 paper") [1].
> * To our knowledge, our work is the first to show a phenomenon like "considering the opposite" from human psychology (which is able to reduce human overconfidence) can improve calibration in LLMs (via top-k verbalized methods).
>
> We hope these points address your concerns regarding the novelty of our contributions.
>
> [1] GPT-4 Technical Report. OpenAI, 2023. arXiv.

---

### Official Review · Reviewer_qYcs · 2023-08-16

**Soundness:** 4

**Excitement:**

4: Strong: This paper deepens the understanding of some phenomenon or lowers the barriers to an existing research direction.

**Paper Topic And Main Contributions:**

In this paper the authors present an evaluation of the calibration of RLHF fine-tuned LLMs. In order to do this, they extract different sources of confidence scores from the models and later apply a suite of metrics to measure how the confidence scores relate to the likelihood that the answer is correct. Their main conclusions are that verbalized confidences are better calibrated than using conditional distributions as confidence scores, as this ones tend to be overconfident. Verbalized confidences token level outputs by the language models. It is also interesting to see how generating multiple hypotheses improves calibration.

**Questions For The Authors:**

- How much do you think that potential contamination could affect the models overconfidence?
- Do you plan to compare these numbers with the non RLHF fine tuned models?

**Reasons To Accept:**

- The paper is well written and it’s easy to follow.
- The topic of the paper is very relevant due to the broad deployment of RLHF models.

**Reasons To Reject:**

- All the LMs analysed are closed so it’s difficult to know how potential data leakage affects the conclusions.
- They don’t compare the calibration results of the non RLHF models.

**Reproducibility:**

4: Could mostly reproduce the results, but there may be some variation because of sample variance or minor variations in their interpretation of the protocol or method.

**Reviewer Confidence:**

4: Quite sure. I tried to check the important points carefully. It's unlikely, though conceivable, that I missed something that should affect my ratings.

---

> ### Author Rebuttal · Authors · 2023-08-28
>
> Thank you for your feedback on our work, and your appreciation of its relevance in light of the widespread use of RLHF-LMs!
>
> Regarding data leakage, while it's possible that data leakage might contribute to the overconfidence of RLHF-LMs, we suggest that data leakage does not completely explain the poor performance of the model logits. Past work has shown that overconfidence due to overfitting to the training distribution may be largely mitigated by simple temperature scaling in both image classifiers [1] and large language models [2]. However, our results show that even with temperature scaling, the logits of RLHF-LMs show relatively poor calibration, suggesting that something other than data leakage is involved (likely the RLHF objective, to some extent).
>
> Moreover, in light of the Llama-2 models released since the submission deadline, we have been able to study larger-sized open RLHF-LMs as well. We confirm that the post-RLHF Llama-2 logprobs also have much worse calibration (ECE-t 0.134, 0.207, 0.382) than the pre-RLHF Llama-2 base model (ECE-t 0.032, 0.061, 0.228), following the same pattern as GPT4 [3].
>
> Regarding calibration of non-RLHF models, the emphasis of our study is RLHF-LMs because they represent the wide majority of generative language models used in practice, and previous work has studied calibration of non-RLHF-LMs through fine-tuning [2] or prompting alone [4]. Our study is the first to show that verbalization without any fine-tuning is an effective strategy for improving calibration in RLHF-LMs, adding a significant caveat to the conclusion from [3] (the "GPT-4 paper") that RLHF largely destroys calibration.
>
> Nonetheless, we conduct an additional study of some of the calibration methods for non-RLHF models including the pre-RLHF InstructGPT model and the base Llama-2 70B model. The results are below:
>
> * InstructGPT's model probabilities are well-calibrated across TriviaQA, SciQ, and TruthfulQA (ECE-t 0.040, 0.075, 0.216) in comparison to post-RLHF GPT-3.5/ChatGPT (ECE-t 0.097, 0.180, 0.317). However, InstructGPT demonstrates poor verbalized calibration with k=1 (ECE-t 0.221, 0.264, 0.371) in comparison to GPT-3.5 (ECE-t 0.076, 0.084, 0.256) and struggles to express k=4 verbalized calibration at all, while GPT-3.5 benefits from k=4 verbalization (ECE-t 0.057, 0.051, 0.189), outperforming the average ECE of InstructGPT's logprobs.
>
> * To consider the pre-RLHF Llama-2 base model, we use a few shot prompt that provides example answers and their corresponding model conditional probabilities. Similarly to InstructGPT, Llama-2-base has well-calibrated model conditional probabilities across three datasets (ECE-t 0.032, 0.061, 0.228) but has uninformative k=1 verbalized calibration (ECE-t 0.129, 0.245, 0.320).
>
>
> Although future investigation of more complex methods may be able to elicit calibrated verbalized confidences from pre-RLHF models, this ability is not easily found in pre-RLHF models.
>
> [1] On Calibration of Modern Neural Networks. Chuan Guo et al., 2017. ICML.
>
> [2] Teaching models to express their uncertainty in words. Stephanie Lin et al., 2022. TMLR.
>
> [3] GPT-4 Technical Report. OpenAI, 2023. arXiv.
>
> [4] Language Models (Mostly) Know What They Know. Saurav Kadavath et al., 2022. arXiv.

---

### Meta-Review · Area_Chair_Pwwf · 2023-09-18

**Recommendation:** 4

**Metareview:**

This paper studies model calibration of LLMs trained with RLHF based on existing observation that RLHF worsens the calibration of model, i.e. the confidence of model predictions less correlates with accuracy. Due to lack of log probabilities of closed models, authors propose to use verbalized probabilities. They also show that prompting a model to produce several answer choices before giving its confidence scores significantly improves calibration of verbalized probabilities. Per reviewers' requests, authors added experiments with llama2 and llama2-RLHF models. I think this paper studies a very important issue -- model calibration in the LLM regime, and provides timely and interesting studies regarding how to measure model calibration for closed models. I encourage authors to add more results of open-sourced models in the camera-ready and include more evidence on "RLHF worsens the calibration".

---

### Decision · Program_Chairs · 2023-10-07

**Decision:**

Accept-Main

**Comment:**

This paper studies model calibration of LLMs trained with RLHF based on existing observation that RLHF worsens the calibration of model, i.e. the confidence of model predictions less correlates with accuracy. Due to lack of log probabilities of closed models, authors propose to use verbalized probabilities. They also show that prompting a model to produce several answer choices before giving its confidence scores significantly improves calibration of verbalized probabilities. Per reviewers' requests, authors added experiments with llama2 and llama2-RLHF models. I think this paper studies a very important issue -- model calibration in the LLM regime, and provides timely and interesting studies regarding how to measure model calibration for closed models. I encourage authors to add more results of open-sourced models in the camera-ready and include more evidence on "RLHF worsens the calibration".